# Generative Image Modeling Using Spatial LSTMs

**Lucas Theis**
University of Tübingen
72076 Tübingen, Germany
lucas@bethgelab.org

**Matthias Bethge**
University of Tübingen
72076 Tübingen, Germany
matthias@bethgelab.org

## Abstract

Modeling the distribution of natural images is challenging, partly because of strong statistical dependencies which can extend over hundreds of pixels. Recurrent neural networks have been successful in capturing long-range dependencies in a number of problems but only recently have found their way into generative image models. We here introduce a recurrent image model based on multi-dimensional long short-term memory units which are particularly suited for image modeling due to their spatial structure. Our model scales to images of arbitrary size and its likelihood is computationally tractable. We find that it outperforms the state of the art in quantitative comparisons on several image datasets and produces promising results when used for texture synthesis and inpainting.

## 1   Introduction

The last few years have seen tremendous progress in learning useful image representations [6]. While early successes were often achieved through the use of generative models [e.g., 13, 23, 30], recent breakthroughs were mainly driven by improvements in supervised techniques [e.g., 20, 34]. Yet unsupervised learning has the potential to tap into the much larger source of unlabeled data, which may be important for training bigger systems capable of a more general scene understanding. For example, multimodal data is abundant but often unlabeled, yet can still greatly benefit unsupervised approaches [36].

Generative models provide a principled approach to unsupervised learning. A perfect model of natural images would be able to optimally predict parts of an image given other parts of an image and thereby clearly demonstrate a form of scene understanding. When extended by labels, the Bayesian framework can be used to perform semi-supervised learning in the generative model [19, 28] while it is less clear how to combine other unsupervised approaches with discriminative learning. Generative image models are also useful in more traditional applications such as image reconstruction [33, 35, 49] or compression [47].

Recently there has been a renewed strong interest in the development of generative image models [e.g., 4, 8, 10, 11, 18, 24, 31, 35, 45, 47]. Most of this work has tried to bring to bear the flexibility of deep neural networks on the problem of modeling the distribution of natural images. One challenge in this endeavor is to find the right balance between tractability and flexibility. The present article contributes to this line of research by introducing a fully tractable yet highly flexible image model.

Our model combines multi-dimensional recurrent neural networks [9] with mixtures of experts. More specifically, the backbone of our model is formed by a spatial variant of *long short-term memory* (LSTM) [14]. One-dimensional LSTMs have been particularly successful in modeling text and speech [e.g., 38, 39], but have also been used to model the progression of frames in video [36] and very recently to model single images [11]. In contrast to earlier work on modeling images, here we use *multi-dimensional LSTMs* [9] which naturally lend themselves to the task of generative image modeling due to their spatial structure and ability to capture long-range correlations.

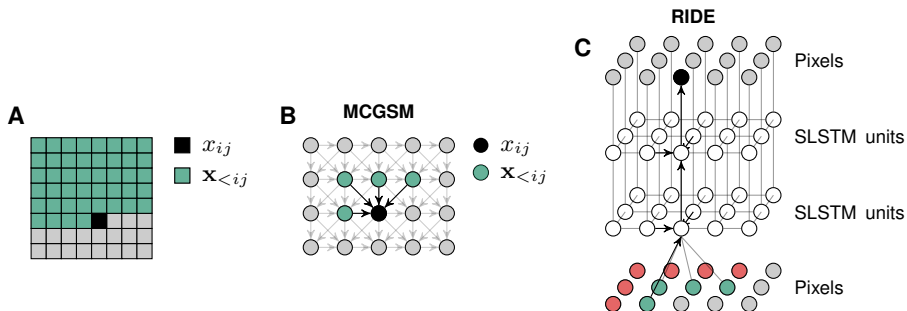

Figure 1: (**A**) We factorize the distribution of images such that the prediction of a pixel (black) may depend on any pixel in the upper-left green region. (**B**) A graphical model representation of an MCGSM with a causal neighborhood limited to a small region. (**C**) A visualization of our recurrent image model with two layers of spatial LSTMs. The pixels of the image are represented twice and some arrows are omitted for clarity. Through feedforward connections, the prediction of a pixel depends directly on its neighborhood (green), but through recurrent connections it has access to the information in a much larger region (red).

To model the distribution of pixels conditioned on the hidden states of the neural network, we use *mixtures of conditional Gaussian scale mixtures* (MCGSMs) [41]. This class of models can be viewed as a generalization of Gaussian mixture models, but their parametrization makes them much more suitable for natural images. By treating images as instances of a stationary stochastic process, this model allows us to sample and capture the correlations of arbitrarily large images.

## 2   A recurrent model of natural images

In the following, we first review and extend the MCGSM [41] and multi-dimensional LSTMs [9] before explaining how to combine them into a recurrent image model. Section 3 will demonstrate the validity of our approach by evaluating and comparing the model on a number of image datasets.

### 2.1   Factorized mixtures of conditional Gaussian scale mixtures

One successful approach to building flexible yet tractable generative models has been to use fully-visible belief networks [21, 27]. To apply such a model to images, we have to give the pixels an ordering and specify the distribution of each pixel conditioned on its parent pixels. Several parametrizations have been suggested for the conditional distributions in the context of natural images [5, 15, 41, 44, 45]. We here review and extend the work of Theis et al. [41] who proposed to use *mixtures of conditional Gaussian scale mixtures* (MCGSMs).

Let $\mathbf{x}$ be a grayscale image patch and $x_{ij}$ be the intensity of the pixel at location $ij$. Further, let $\mathbf{x}_{<ij}$ designate the set of pixels $x_{mn}$ such that $m < i$ or $m = i$ and $n < j$ (Figure 1A). Then

$$p(\mathbf{x}; \boldsymbol{\theta}) = \prod_{i,j} p(x_{ij} \mid \mathbf{x}_{<ij}; \boldsymbol{\theta}) \tag{1}$$

for the distribution of any parametric model with parameters $\boldsymbol{\theta}$. Note that this factorization does not make any independence assumptions but is simply an application of the probability chain rule. Further note that the conditional distributions all share the same set of parameters. One way to improve the representational power of a model is thus to endow each conditional distribution with its own set of parameters,

$$p(\mathbf{x}; \{\boldsymbol{\theta}_{ij}\}) = \prod_{i,j} p(x_{ij} \mid \mathbf{x}_{<ij}; \boldsymbol{\theta}_{ij}). \tag{2}$$

Applying this trick to mixtures of Gaussian scale mixtures (MoGSMs) yields the MCGSM [40]. Untying shared parameters can drastically increase the number of parameters. For images, it can easily be reduced again by adding assumptions. For example, we can limit $\mathbf{x}_{<ij}$ to a smaller neighborhood surrounding the pixel by making a Markov assumption. We will refer to the resulting set of parents as the pixel's *causal neighborhood* (Figure 1B). Another reasonable assumption is stationarity or shift invariance, in which case we only have to learn one set of parameters $\boldsymbol{\theta}_{ij}$ which can then

be used at every pixel location. Similar to convolutions in neural networks, this allows the model to easily scale to images of arbitrary size. While this assumption reintroduces parameter sharing constraints into the model, the constraints are different from the ones induced by the joint mixture model.

The conditional distribution in an MCGSM takes the form of a mixture of experts,

$$p(x_{ij} \mid \mathbf{x}_{<ij}, \boldsymbol{\theta}_{ij}) = \sum_{c,s} \underbrace{p(c, s \mid \mathbf{x}_{<ij}, \boldsymbol{\theta}_{ij})}_{\text{gate}} \underbrace{p(x_{ij} \mid \mathbf{x}_{<ij}, c, s, \boldsymbol{\theta}_{ij})}_{\text{expert}}, \tag{3}$$

where the sum is over mixture component indices $c$ corresponding to different covariances and scales $s$ corresponding to different variances. The gates and experts in an MCGSM are given by

$$p(c, s \mid \mathbf{x}_{<ij}) \propto \exp\left(\eta_{cs} - \tfrac{1}{2} e^{\alpha_{cs}} \mathbf{x}_{<ij}^{\top} \mathbf{K}_c \mathbf{x}_{<ij}\right), \tag{4}$$

$$p(x_{ij} \mid \mathbf{x}_{<ij}, c, s) = \mathcal{N}(x_{ij}; \mathbf{a}_c^{\top} \mathbf{x}_{<ij}, e^{-\alpha_{cs}}), \tag{5}$$

where $\mathbf{K}_c$ is positive definite. The number of parameters of an MCGSM still grows quadratically with the dimensionality of the causal neighborhood. To further reduce the number of parameters, we introduce a factorized form of the MCGSM with additional parameter sharing by replacing $\mathbf{K}_c$ with $\sum_n \beta_{cn}^2 \mathbf{b}_n \mathbf{b}_n^{\top}$. This *factorized MCGSM* allows us to use larger neighborhoods and more mixture components. A detailed derivation of a more general version which also allows for multivariate pixels is given in Supplementary Section 1.

## 2.2 Spatial long short-term memory

In the following we briefly describe the *spatial LSTM* (SLSTM), a special case of the multi-dimensional LSTM first described by Graves & Schmidhuber [9]. At the core of the model are memory units $\mathbf{c}_{ij}$ and hidden units $\mathbf{h}_{ij}$. For each location $ij$ on a two-dimensional grid, the operations performed by the spatial LSTM are given by

$$\mathbf{c}_{ij} = \mathbf{g}_{ij} \odot \mathbf{i}_{ij} + \mathbf{c}_{i,j-1} \odot \mathbf{f}_{ij}^c + \mathbf{c}_{i-1,j} \odot \mathbf{f}_{ij}^r, \qquad \begin{pmatrix} \mathbf{g}_{ij} \\ \mathbf{o}_{ij} \\ \mathbf{i}_{ij} \\ \mathbf{f}_{ij}^r \\ \mathbf{f}_{ij}^c \end{pmatrix} = \begin{pmatrix} \tanh \\ \sigma \\ \sigma \\ \sigma \\ \sigma \end{pmatrix} T_{\mathbf{A},\mathbf{b}} \begin{pmatrix} \mathbf{x}_{<ij} \\ \mathbf{h}_{i,j-1} \\ \mathbf{h}_{i-1,j} \end{pmatrix}, \tag{6}$$
$$\mathbf{h}_{ij} = \tanh\left(\mathbf{c}_{ij} \odot \mathbf{o}_{ij}\right),$$

where $\sigma$ is the logistic sigmoid function, $\odot$ indicates a pointwise product, and $T_{\mathbf{A},\mathbf{b}}$ is an affine transformation which depends on the only parameters of the network $\mathbf{A}$ and $\mathbf{b}$. The gating units $\mathbf{i}_{ij}$ and $\mathbf{o}_{ij}$ determine which memory units are affected by the inputs through $\mathbf{g}_{ij}$, and which memory states are written to the hidden units $\mathbf{h}_{ij}$. In contrast to a regular LSTM defined over time, each memory unit of a spatial LSTM has two preceding states $\mathbf{c}_{i,j-1}$ and $\mathbf{c}_{i-1,j}$ and two corresponding forget gates $\mathbf{f}_{ij}^c$ and $\mathbf{f}_{ij}^r$.

## 2.3 Recurrent image density estimator

We use a grid of SLSTM units to sequentially read relatively small neighborhoods of pixels from the image, producing a hidden vector at every pixel. The hidden states are then fed into a factorized MCGSM to predict the state of the corresponding pixel, that is, $p(x_{ij} \mid \mathbf{x}_{<ij}) = p(x_{ij} \mid \mathbf{h}_{ij})$. Importantly, the state of the hidden vector only depends on pixels in $\mathbf{x}_{<ij}$ and does not violate the factorization given in Equation 1. Nevertheless, the recurrent network allows this *recurrent image density estimator* (RIDE) to use pixels of a much larger region for prediction, and to nonlinearly transform the pixels before applying the MCGSM. We can further increase the representational power of the model by stacking spatial LSTMs to obtain a deep yet still completely tractable recurrent image model (Figure 1C).

## 2.4 Related work

Larochelle & Murray [21] derived a tractable density estimator (NADE) in a manner similar to how the MCGSM was derived [41], but using restricted Boltzmann machines (RBM) instead of mixture models as a starting point. In contrast to the MCGSM, NADE tries to keep the weight sharing

constraints induced by the RBM (Equation 1). Uria et al. extended NADE to real values [44] and introduced hidden layers to the model [45]. Gregor et al. [10] describe a related autoregressive network for binary data which additionally allows for stochastic hidden units.

Gregor et al. [11] used one-dimensional LSTMs to generate images in a sequential manner (DRAW). Because the model was defined over Bernoulli variables, normalized RGB values had to be treated as probabilities, making a direct comparison with other image models difficult. In contrast to our model, the presence of stochastic latent variables in DRAW means that its likelihood cannot be evaluated but has to be approximated.

Ranzato et al. [31] and Srivastava et al. [37] use one-dimensional recurrent neural networks to model videos, but recurrency is not used to describe the distribution over individual frames. Srivastava et al. [37] optimize a squared error corresponding to a Gaussian assumption, while Ranzato et al. [31] try to side-step having to model pixel intensities by quantizing image patches. In contrast, here we also try to solve the problem of modeling pixel intensities by using an MCGSM, which is equipped to model heavy-tailed as well as multi-modal distributions.

## 3  Experiments

RIDE was trained using stochastic gradient descent with a batch size of 50, momentum of 0.9, and a decreasing learning rate varying between 1 and $10^{-4}$. After each pass through the training set, the MCGSM of RIDE was finetuned using L-BFGS for up to 500 iterations before decreasing the learning rate. No regularization was used except for early stopping based on a validation set. Except where indicated otherwise, the recurrent model used a 5 pixel wide neighborhood and an MCGSM with 32 components and 32 quadratic features ($\mathbf{b}_n$ in Section 2.1). Spatial LSTMs were implemented using the Caffe framework [17]. Where appropriate, we augmented the data by horizontal or vertical flipping of images.

We found that conditionally whitening the data greatly sped up the training process of both models. Letting $\mathbf{y}$ represent a pixel and $\mathbf{x}$ its causal neighborhood, conditional whitening replaces these with

$$\hat{\mathbf{x}} = \mathbf{C}_{\mathbf{xx}}^{-\frac{1}{2}}\left(\mathbf{x} - \mathbf{m}_{\mathbf{x}}\right), \quad \hat{\mathbf{y}} = \mathbf{W}(\mathbf{y} - \mathbf{C}_{\mathbf{yx}}\mathbf{C}_{\mathbf{xx}}^{-\frac{1}{2}}\hat{\mathbf{x}} - \mathbf{m}_{\mathbf{y}}), \quad \mathbf{W} = (\mathbf{C}_{\mathbf{yy}} - \mathbf{C}_{\mathbf{yx}}\mathbf{C}_{\mathbf{xx}}^{-1}\mathbf{C}_{\mathbf{yx}}^{\top})^{-\frac{1}{2}}, \quad (7)$$

where $\mathbf{C}_{\mathbf{yx}}$ is the covariance of $\mathbf{y}$ and $\mathbf{x}$, and $\mathbf{m}_{\mathbf{x}}$ is the mean of $\mathbf{x}$. In addition to speeding up training, this variance normalization step helps to make the learning rates less dependent on the training data. When evaluating the conditional log-likelihood, we compensate for the change in variance by adding the log-Jacobian $\log|\det \mathbf{W}|$. Note that this preconditioning introduces a shortcut connection from the pixel neighborhood to the predicted pixel which is not shown in Figure 1C.

### 3.1  Ensembles

Uria et al. [45] found that forming ensembles of their autoregressive model over different pixel orderings significantly improved performance. We here consider a simple trick to produce an ensemble without the need for training different models or to change training procedures. If $\mathbf{T}_k$ are linear transformations leaving the targeted image distribution invariant (or approximately invariant) and if $p$ is the distribution of a pretrained model, then we form the ensemble $\frac{1}{K}\sum_k p(\mathbf{T}_k\mathbf{x})|\det \mathbf{T}_k|$. Note that this is simply a mixture model over images $\mathbf{x}$. We considered rotating as well as flipping images along the horizontal and vertical axes (yielding an ensemble over 8 transformations). While it could be argued that most of these transformations do not leave the distribution over natural images invariant, we nevertheless observed a noticeable boost in performance.

### 3.2  Natural images

Several recent image models have been evaluated on small image patches sampled from the Berkeley segmentation dataset (BSDS300) [25]. Although our model's strength lies in its ability to scale to large images and to capture long-range correlations, we include results on BSDS300 to make a connection to this part of the literature. We followed the protocol of Uria et al. [44]. The RGB images were turned to grayscale, uniform noise was added to account for the integer discretization, and the resulting values were divided by 256. The training set of 200 images was split into 180 images for training and 20 images for validation, while the test set contained 100 images. We

| Model | 63 dim. [nat] | 64 dim. [bit/px] | ∞ dim. [bit/px] |
|---|---|---|---|
| RNADE [44] | 152.1 | 3.346 | - |
| RNADE, 1 hl [45] | 143.2 | 3.146 | - |
| RNADE, 6 hl [45] | 155.2 | 3.416 | - |
| EoRNADE, 6 layers [45] | 157.0 | 3.457 | - |
| GMM, 200 comp. [47, 50] | 153.7 | 3.360 | - |
| STM, 200 comp. [46] | 155.3 | 3.418 | - |
| Deep GMM, 3 layers [47] | 156.2 | 3.439 | - |
| MCGSM, 16 comp. | 155.1 | 3.413 | 3.688 |
| MCGSM, 32 comp. | 155.8 | 3.430 | 3.706 |
| MCGSM, 64 comp. | 156.2 | 3.439 | 3.716 |
| MCGSM, 128 comp. | 156.4 | 3.443 | 3.717 |
| EoMCGSM, 128 comp. | **158.1** | **3.481** | 3.748 |
| RIDE, 1 layer | 150.7 | 3.293 | 3.802 |
| RIDE, 2 layers | 152.1 | 3.346 | 3.869 |
| EoRIDE, 2 layers | 154.5 | 3.400 | **3.899** |

| Model | 256 dim. [bit/px] | ∞ dim. [bit/px] |
|---|---|---|
| GRBM [13] | 0.992 | - |
| ICA [1, 48] | 1.072 | - |
| GSM | 1.349 | - |
| ISA [7, 16] | 1.441 | - |
| MoGSM, 32 comp. [40] | 1.526 | - |
| MCGSM, 32 comp. | 1.615 | 1.759 |
| RIDE, 1 layer, 64 hid. | **1.650** | 1.816 |
| RIDE, 1 layer, 128 hid. | - | 1.830 |
| RIDE, 2 layers, 64 hid. | - | 1.829 |
| RIDE, 2 layers, 128 hid. | - | 1.839 |
| EoRIDE, 2 layers, 128 hid. | - | **1.859** |

Table 1: Average log-likelihoods and log-likelihood rates for image patches (without/with DC comp.) and large images extracted from BSDS300 [25].

Table 2: Average log-likelihood rates for image patches and large images extracted from van Hateren's dataset [48].

extracted 8 by 8 image patches from each set and subtracted the average pixel intensity such that each patch's DC component was zero. Because the resulting image patches live on a 63 dimensional subspace, the bottom-right pixel was discarded. We used $1.6 \cdot 10^6$ patches for training, $1.8 \cdot 10^5$ patches for validation, and $10^6$ test patches for evaluation.

MCGSMs have not been evaluated on this dataset and so we first tested MCGSMs by training a single factorized MCGSM for each pixel conditioned on all previous pixels in a fixed ordering. We find that already an MCGSM (with 128 components and 48 quadratic features) outperforms all single models including a deep Gaussian mixture model [46] (Table 1). Our ensemble of MCGSMs[1] outperforms an ensemble of RNADEs with 6 hidden layers, which to our knowledge is currently the best result reported on this dataset.

Training the recurrent image density estimator (RIDE) on the 63 dimensional dataset is more cumbersome. We tried padding image patches with zeros, which was necessary to be able to compute a hidden state at every pixel. The bottom-right pixel was ignored during training and evaluation. This simple approach led to a reduction in performance relative to the MCGSM (Table 1). A possible explanation is that the model cannot distinguish between pixel intensities which are zero and zeros in the padded region. Supplying the model with additional binary indicators as inputs (one for each neighborhood pixel) did not solve the problem.

However, we found that RIDE outperforms the MCGSM by a large margin when images were treated as instances of a stochastic process (that is, using infinitely large images). MCGSMs were trained for up to 3000 iterations of L-BFGS on $10^6$ pixels and corresponding causal neighborhoods extracted from the training images. Causal neighborhoods were 9 pixels wide and 5 pixels high. RIDE was trained for 8 epochs on image patches of increasing size ranging from 8 by 8 to 22 by 22 pixels (that is, gradients were approximated as in backpropagation through time [32]). The right column in Table 1 shows average log-likelihood rates for both models. Analogously to the entropy rate [3], we have for the expected log-likelihood rate:

$$\lim_{N \to \infty} \mathbb{E}\left[\log p(\mathbf{x})/N^2\right] = \mathbb{E}[\log p(x_{ij} \mid \mathbf{x}_{<ij})], \tag{8}$$

where $\mathbf{x}$ is an $N$ by $N$ image patch. An average log-likelihood rate can be directly computed for the MCGSM, while for RIDE and ensembles we approximated it by splitting the test images into 64 by 64 patches and evaluating on those.

To make the two sets of numbers more comparable, we transformed nats as commonly reported on the 63 dimensional data, $\ell_{1:63}$, into a bit per pixel log-likelihood rate using the formula $(\ell_{1:63} + \ell_{DC} + \ln |\det \mathbf{A}|)/64/\ln(2)$. This takes into account a log-likelihood for the missing DC component,

| Model | [bit/px] |
|---|---|
| MCGSM, 12 comp. [41] | 1.244 |
| MCGSM, 32 comp. | 1.294 |
| Diffusion [35] | 1.489 |
| RIDE, 64 hid., 1 layer | 1.402 |
| RIDE, 64 hid., 1 layer, ext. | 1.416 |
| RIDE, 64 hid., 2 layers | 1.438 |
| RIDE, 64 hid., 3 layers | 1.454 |
| RIDE, 128 hid., 3 layers | 1.489 |
| EoRIDE, 128 hid., 3 layers | **1.501** |

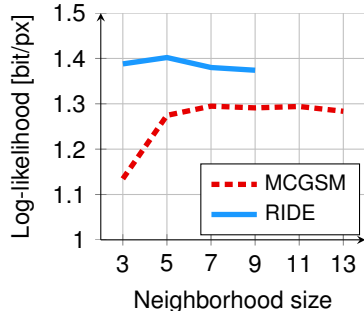

Table 3: Average log-likelihood rates on dead leaf images. A deep recurrent image model is on a par with a deep diffusion model [35]. Using ensembles we are able to further improve the likelihood.

Figure 2: Model performance on dead leaves as a function of the causal neighborhood width. Simply increasing the neighborhood size of the MCGSM is not sufficient to improve performance.

$\ell_{DC} = 0.5020$, and the Jacobian of the transformations applied during preprocessing, $\ln|\det \mathbf{A}| = -4.1589$ (see Supplementary Section 2.2 for details). The two rates in Table 1 are comparable in the sense that their differences express how much better one model would be at losslessly compressing BSDS300 test images than another, where patch-based models would compress patches of an image independently. We highlighted the best result achieved with each model in gray. Note that most models in this list do not scale as well to large images as the MCGSM or RIDE (GMMs in particular) and are therefore unlikely to benefit as much from increasing the patch size.

A comparison of the log-likelihood rates reveals that an MCGSM with 16 components applied to large images already captures more correlations than any model applied to small image patches. The difference is particularly striking given that the factorized MCGSM has approximately 3,000 parameters while a GMM with 200 components has approximately 400,000 parameters. Using an ensemble of RIDEs, we are able to further improve this number significantly (Table 1).

Another dataset frequently used to test generative image models is the dataset published by van Hateren and van der Schaaf [48]. Details of the preprocessing used in this paper are given in Supplementary Section 3. We reevaluated several models for which the likelihood has been reported on this dataset [7, 40, 41, 42]. Likelihood rates as well as results on 16 by 16 patches are given in Table 2. Because of the larger patch size, RIDE here already outperforms the MCGSM on patches.

### 3.3 Dead leaves

Dead leaf images are generated by superimposing disks of random intensity and size on top of each other [22, 26]. This simple procedure leads to images which already share many of the statistical properties and challenges of natural images, such as occlusions and long-range correlations, while leaving out others such as non-stationary statistics. They therefore provide an interesting test case for natural image models.

We used a set of 1,000 images, where each image is 256 by 256 pixels in size. We compare the performance of RIDE to the MCGSM and a very recently introduced deep multiscale model based on a diffusion process [35]. The same 100 images as in previous literature [35, 41] were used for evaluation and we used the remaining images for training. We find that the introduction of an SLSTM with 64 hidden units greatly improves the performance of the MCGSM. We also tried an extended version of the SLSTM which included memory units as additional inputs (right-hand side of Equation 6). This yielded a small improvement in performance (5th row in Table 3) while adding layers or using more hidden units led to more drastic improvements. Using 3 layers with 128 hidden units in each layer, we find that our recurrent image model is on a par with the deep diffusion model. By using ensembles, we are able to beat all previously published results for this dataset (Table 3).

Figure 2 shows that the improved performance of RIDE is not simply due to an effectively larger causal neighborhood but that the nonlinear transformations performed by the SLSTM units matter. Simply increasing the neighborhood size of an MCGSM does not yield the same improvement. Instead, the performance quickly saturates. We also find that the performance of RIDE slightly deteriorates with larger neighborhoods, which is likely caused by optimization difficulties.

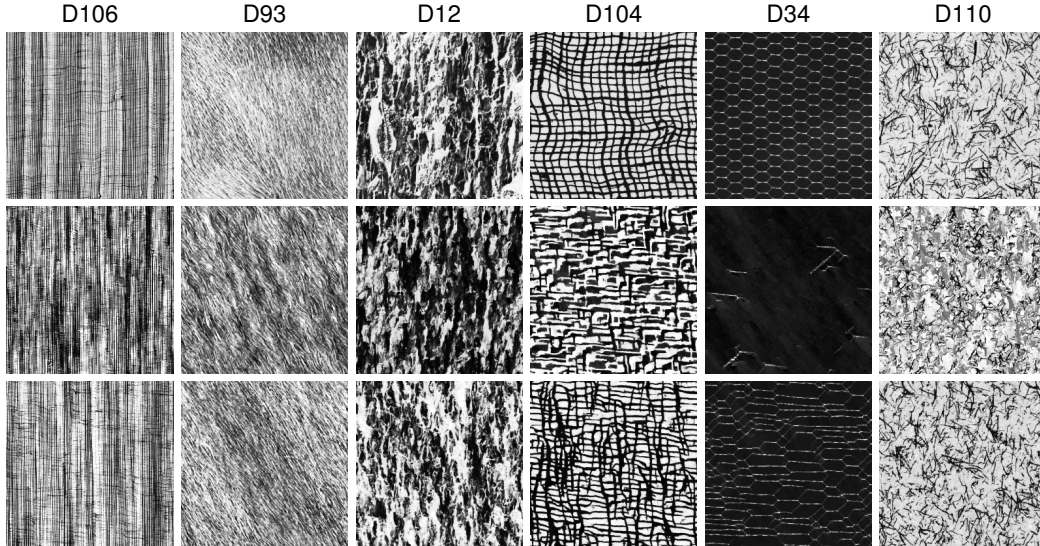

Figure 3: From top to bottom: A 256 by 256 pixel crop of a texture [2], a sample generated by an MCGSM trained on the full texture [7], and a sample generated by RIDE. This illustrates that our model can capture a variety of different statistical patterns. The addition of the recurrent neural network seems particularly helpful where there are strong long-range correlations (D104, D34).

## 3.4 Texture synthesis and inpainting

To get an intuition for the kinds of correlations which RIDE can capture or fails to capture, we tried to use it to synthesize textures. We used several 640 by 640 pixel textures published by Brodatz [2]. The textures were split into sixteen 160 by 160 pixel regions of which 15 were used for training and one randomly selected region was kept for testing purposes. RIDE was trained for up to 6 epochs on patches of increasing size ranging from 20 by 20 to 40 by 40 pixels.

Samples generated by an MCGSM and RIDE are shown in Figure 3. Both models are able to capture a wide range of correlation structures. However, the MCGSM seems to struggle with textures having bimodal marginal distributions and periodic patterns (D104, D34, and D110). RIDE clearly improves on these textures, although it also struggles to faithfully reproduce periodic structure. Possible explanations include that LSTMs are not well suited to capture periodicities, or that these failures are not penalized strong enough by the likelihood. For some textures, RIDE produces samples which are nearly indistinguishable from the real textures (D106 and D110).

One application of generative image models is inpainting [e.g., 12, 33, 35]. As a proof of concept, we used our model to inpaint a large (here, 71 by 71 pixels) region in textures (Figure 4). Missing pixels were replaced by sampling from the posterior of RIDE. Unlike the joint distribution, the posterior distribution cannot be sampled directly and we had to resort to Markov chain Monte Carlo methods. We found the following *Metropolis within Gibbs* [43] procedure to be efficient enough. The missing pixels were initialized via ancestral sampling. Since ancestral sampling is cheap, we generated 5 candidates and used the one with the largest posterior density. Following initialization, we sequentially updated overlapping 5 by 5 pixel regions via Metropolis sampling. Proposals were generated via ancestral sampling and accepted using the acceptance probability

$$\alpha = \min\left\{1, \frac{p(\mathbf{x}')}{p(\mathbf{x})}\frac{p(\mathbf{x}_{ij}|\mathbf{x}_{<ij})}{p(\mathbf{x}'_{ij}|\mathbf{x}_{<ij})}\right\}, \tag{9}$$

where here $\mathbf{x}_{ij}$ represents a 5 by 5 pixel patch and $\mathbf{x}'_{ij}$ its proposed replacement. Since evaluating the joint and conditional densities on the entire image is costly, we approximated $p$ using RIDE applied to a 19 by 19 pixel patch surrounding $ij$. Randomly flipping images vertically or horizontally in between the sampling further helped. Figure 4 shows results after 100 Gibbs sampling sweeps.

## 4 Conclusion

We have introduced RIDE, a deep but tractable recurrent image model based on spatial LSTMs. The model exemplifies how recent insights in deep learning can be exploited for generative image

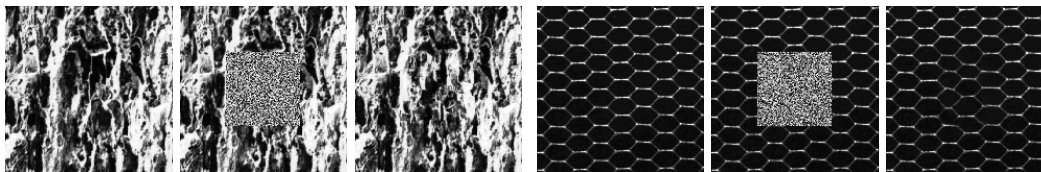

Figure 4: The center portion of a texture (left and center) was reconstructed by sampling from the posterior distribution of RIDE (right).

modeling and shows superior performance in quantitative comparisons. RIDE is able to capture many different statistical patterns, as demonstrated through its application to textures. This is an important property considering that on an intermediate level of abstraction natural images can be viewed as collections of textures.

We have furthermore introduced a factorized version of the MCGSM which allowed us to use more experts and larger causal neighborhoods. This model has few parameters, is easy to train and already on its own performs very well as an image model. It is therefore an ideal building block and may be used to extend other models such as DRAW [11] or video models [31, 37].

Deep generative image models have come a long way since deep belief networks have first been applied to natural images [29]. Unlike convolutional neural networks in object recognition, however, no approach has as of yet proven to be a likely solution to the problem of generative image modeling. Further conceptual work will be necessary to come up with a model which can handle both the more abstract high-level as well as the low-level statistics of natural images.

## Acknowledgments

The authors would like to thank Aäron van den Oord for insightful discussions and Wieland Brendel, Christian Behrens, and Matthias Kümmerer for helpful input on this paper. This study was financially supported by the German Research Foundation (DFG; priority program 1527, BE 3848/2-1).

## Footnotes

[1] Details on how the ensemble of transformations can be applied despite the missing bottom-right pixel are given in Supplementary Section 2.1.

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
