[Supplementary Material]

# 1 Factorized mixtures of conditional Gaussian scale mixtures

The conditional distribution of a mixture of Gaussian scale mixtures (GSMs) over one set of pixels $\mathbf{y}$ given a disjoint set of pixels $\mathbf{x}$ takes the form [5]

$$p(\mathbf{y} \mid \mathbf{x}) = \sum_{cs} p(c, s \mid \mathbf{x}) p(\mathbf{y} \mid \mathbf{x}, c, s), \tag{1}$$

$$p(c, s \mid \mathbf{x}) \propto |\lambda_{cs} \mathbf{K}_c|^{\frac{1}{2}} \exp\left(-\frac{1}{2} \lambda_{cs} \mathbf{x}^\top \mathbf{K}_c \mathbf{x}\right), \tag{2}$$

$$p(\mathbf{y} \mid \mathbf{x}, c, s) = |\mathbf{M}_c|^{\frac{1}{2}} \exp\left(-\frac{1}{2} \lambda_{cs} (\mathbf{y} - \mathbf{A}_c \mathbf{x})^\top \mathbf{M}_c (\mathbf{y} - \mathbf{A}_c \mathbf{x})\right) / (2\pi)^{\frac{D}{2}}, \tag{3}$$

where $\lambda_{cs} > 0$ and $\mathbf{M}_c$ and $\mathbf{K}_c$ are positive definite. This assumes that the means of the GSMs are zero and the weights of the components are equal. We here consider the more general form

$$p(c, s \mid \mathbf{x}) \propto \pi_{cs} |\lambda_{cs} \mathbf{K}_c|^{\frac{1}{2}} \exp\left(-\frac{1}{2} \lambda_{cs} (\mathbf{x} - \mathbf{u}_c)^\top \mathbf{K}_c (\mathbf{x} - \mathbf{u}_c)\right), \tag{4}$$

$$p(\mathbf{y} \mid \mathbf{x}, c, s) = |\mathbf{M}_c|^{\frac{1}{2}} \exp\left(-\frac{1}{2} \lambda_{cs} (\mathbf{y} - \mathbf{A}_c \mathbf{x} - \mathbf{m}_c)^\top \mathbf{M}_c (\mathbf{y} - \mathbf{A}_c \mathbf{x} - \mathbf{m}_c)\right) / (2\pi)^{\frac{D}{2}}, \tag{5}$$

with $\pi_{cs} > 0$. This parametrization has two problems. First, the positivity constraints make this parametrization unsuitable for stochastic gradient descent. We can easily solve this problem by reparametrization. Second, the number of parameters grows quadratically with the dimensionality of $\mathbf{x}$. If $\mathbf{x}$ represents a causal neighborhood, then the number of parameters grows quartically with the width of the neighborhood. We therefore replace the matrices $\mathbf{K}_c$ with low-rank approximations,

$$\mathbf{K}_c = \sum_n \beta_{cn}^2 \mathbf{b}_n \mathbf{b}_n^\top, \tag{6}$$

using rank one basis matrices $\mathbf{b}_n \mathbf{b}_n^\top$ which are furthermore shared between the components. The squaring of the weights $\beta_{cn}$ ensures that the $\mathbf{K}_c$ stay positive semi-definite. This constraint may be dropped in the MCGSM, but we did not explore this option. After reparametrizing, we obtain

$$p(c, s \mid \mathbf{x}) \propto \exp\left(\eta_{cs} - \frac{1}{2} e^{\alpha_{cs}} \sum_n \beta_{cn}^2 (\mathbf{b}_n^\top \mathbf{x})^2 + e^{\alpha_{cs}} \mathbf{w}_c^\top \mathbf{x}\right), \tag{7}$$

$$p(\mathbf{y} \mid \mathbf{x}, c, s) = |\mathbf{L}_c| \exp\left(\frac{M}{2} \alpha_{cs} - \frac{1}{2} e^{\alpha_{cs}} \left(\mathbf{y} - \mathbf{A}_c^\top \mathbf{x} - \mathbf{m}_c\right)^\top \mathbf{L}_c \mathbf{L}_c^\top \left(\mathbf{y} - \mathbf{A}_c^\top \mathbf{x} - \mathbf{m}_c\right)\right) / (2\pi)^{\frac{1}{2}}, \tag{8}$$

where $\mathbf{L}_c$ is a lower-triangular matrix. We found empirically that the parameters $\mathbf{m}_c$ and $\mathbf{w}_c$ do not make a significant difference for quantitative performance and therefore set them to zero.

# 2 BSDS300

## 2.1 Ensembles

We rotate and flip the 63 dimensional data points $\mathbf{z}$ by first reconstructing the 8 by 8 image patches, applying the transformation, and finally removing the bottom-right pixel again. This can be formalized as

$$\mathbf{Tz} = \mathbf{C} \mathbf{P}_\sigma \mathbf{R} \mathbf{x} = \begin{pmatrix} & 0 \\ \mathbf{I} & \vdots \\ & 0 \end{pmatrix} (\delta_{\sigma(i)j})_{ij} \begin{pmatrix} & \mathbf{I} & \\ -1 & \cdots & -1 \end{pmatrix} \mathbf{z}, \tag{9}$$

where $\sigma$ is a permutation representing rotation or flipping, $P_\sigma$ is its corresponding permutation matrix, $\mathbf{I}$ is a $63 \times 63$ identity matrix, and $\mathbf{C}$ and $\mathbf{R}$ are $63 \times 64$ and $64 \times 63$ dimensional matrices, respectively.

As we will show, transformations of this type are volume preserving, that is, their determinant is 1 and the Jacobian can be ignored when evaluating $p(\mathbf{Tz})|\det \mathbf{T}|$. We have

$$T_{ij} = R_{\sigma(i)j} = \begin{cases} 1 & \text{if } \sigma(i) = j \\ -1 & \text{if } \sigma(i) = 64 \qquad \forall i,j \in \{1,...,63\}. \\ 0 & \text{else.} \end{cases} \tag{10}$$

Using Leibniz formula for determinants,

$$|\det \mathbf{T}| = \left| \sum_{\pi \in S_{63}} \left( \text{sgn}(\pi) \prod_{i=1}^{63} T_{i\pi(i)} \right) \right|, \tag{11}$$

where $S_{63}$ is the set of all permutations over $\{1,...,63\}$. Let $\sigma(k) = 64$ for some $k$. Then for all $i \neq k$ we have $T_{i\pi(i)} = 0$ unless $\pi(i) = \sigma(i)$. Since this constraints at least 62 and therefore all values of the permutation $\pi$, it implies that there is only one permutation $\pi' \in S_{63}$ with a nonzero term in the above expansion. Hence,

$$|\det \mathbf{T}| = \left| \text{sgn}(\pi') \prod_{i=1}^{63} T_{i\pi'(i)} \right| = 1. \tag{12}$$

## 2.2 Log-likelihood rates

Let $\mathbf{x}$ be a vector representation of an 8 by 8 image patch sampled from the BSDS300 dataset before subtraction of the pixel mean. We can think of the removal of the DC component also as a basis transformation,

$$\mathbf{z} = \mathbf{Ax} = \begin{pmatrix} 1-\tau & -\tau & -\tau & \cdots & -\tau \\ -\tau & 1-\tau & -\tau & \cdots & -\tau \\ \vdots & & & & \vdots \\ -\tau & -\tau & \cdots & 1-\tau & -\tau \\ \tau & \tau & \cdots & \tau & \tau \end{pmatrix} \mathbf{x}, \tag{13}$$

where $\tau = 1/64$. Instead of a normalized bottom-right pixel, $z_{64}$ here represents the missing DC component of the image patch. Image patch models are commonly trained on the first 63 dimensions, $\mathbf{z}_{1:63}$. To compute a log-likelihood rate, we extend these models by separately modeling the DC component:

$$p(\mathbf{x}) = q(\mathbf{z}_{1:63})q(z_{64})|\det \mathbf{A}|. \tag{14}$$

Using a histogram with 60 bins we get an average log-likelihood of 0.5020 [nat] for the DC component. The log-Jacobian is -4.1589 [nat], so that the formula

$$(\mathbb{E}\left[\ln q(\mathbf{z}_{1:63})\right] + 0.5020 - 4.1589)/64/\ln(2) \tag{15}$$

can be used to transform average log-likelihoods in nats into log-likelihood rates in bits per pixel.

# 3 Dequantizing van Hateren's dataset

Images of van Hateren and van der Schaaf's dataset are stored using a 16 bit integer representing a linearized grayscale intensity for each pixel [8]. After removing overly blurry images and images containing pixel artefacts, the dataset contains 3632 images of 1024 by 1536 pixels with linearized intensities. We used 3000 images for training and evaluation, and the remaining images for testing. Despite adding uniform noise to account for integer discretization, RIDE discovered quantization artefacts in the data undetected by the other models. This led to unnaturally high likelihoods (on both the training and the test set). To avoid uninteresting solutions, we therefore dequantized images on an individual basis as follows.

For each image, we computed a list of unique pixel values occurring in the image and ordered them, $v_1 < v_2 < \cdots < v_M$. If $x_{ij} = v_m$ and $m < M$, we replaced the pixel with

$$x'_{ij} = v_m + (v_{m+1} - v_m)u_{ij}, \tag{16}$$

where $u_{ij}$ is random noise uniformly distributed between 0 and 1 independently drawn for each pixel. If $m = M$, we used $x'_{ij} = v_M + (v_M - v_{M-1})u_{ij}$. After dequantization, images were log-transformed, $x''_{ij} = \log x'_{ij}$, as is common, and no further preprocessing was applied.

# 4 Samples

Although empirical evidence and theoretical arguments suggest that samples are generally not a good surrogate for generative performance [4, 6, 7], for completeness we here include samples produced by our models.

Images were sampled by initializing boundaries with random white noise and sampling pixels row by row from left to right. We first sampled a slightly larger image and then cropped the pixels at the boundaries to produce a 256 by 256 pixel image. Sampling large images was sometimes unstable, especially for RIDE with multiple layers and when training did not converge. This problem could typically be solved by constraining the conditionally sampled pixel values to a finite range.

Figure 1 shows samples of models trained on BSDS300. RIDE samples look similar to MCGSM samples, but appear to contain stronger long-range correlations. For comparison, we also included samples generated by state-of-the-art Markov random fields (MRFs) [1, 2]. In contrast to our models, the likelihoods of these MRFs is intractable and samples were generated using Markov chain Monte Carlo methods.

Additionally, we also include samples of a model trained on RGB images. RGB pixels were first basis-transformed to separate luminance from color. One two-layer RIDE model was trained on the luminance channel while a second model was trained on the color channels conditioned on the luminance channel. Samples were generated by first sampling the luminance channel, then the two color channels, and then mapping pixels back into RGB space.

Samples for models trained on the dataset of van Hateren and van der Schaaf [8] are shown in Figure 2. In contrast to BSDS300, samples of RIDE here look much more natural and structured than samples generated by an MCGSM.

Despite large improvements in average log-likelihood, samples drawn from RIDE trained on dead leaf images appear visually very similar to samples generated by an MCGSM (Figure 3). Only on closer inspection do RIDE samples appear to contain more disk like objects. For comparison, we also include samples generated by a diffusion model [3].

Figure 1: Samples generated by various models trained on BSDS300. Top row, from left to right: MCGSM, RIDE with 1 SLSTM layer, RIDE with 2 SLSTM layers, gated MRF[1] (mPoT-TConv) [2], and MRF[1] [1]. Bottom row: Additional samples generated by two-layer RIDEs trained on RGB images.

Figure 2: Two samples generated by an MCGSM (left) and two samples generated by a two-layer RIDE (right) trained on van Hateren's dataset [8].

Figure 3: A dead leaf image produced by superimposing disks (left), a sample generated by an MCGSM (center-left), a sample generated by a three-layer RIDE (center-right), and a sample generated by a deep multiscale diffusion model[1] [3] (right). Despite the drastic improvement in likelihood, samples of RIDE appear superficially similar to samples drawn from an MCGSM. The diffusion model is able to produce much larger flat regions, but our results suggest suggest that this is not crucial for good performance in terms of average log-likelihood.

## Footnotes

[1]Samples were generated by and included with permission from authors of the respective papers.