[Reviews · NeurIPS 2015]

Submitted by Assigned_Reviewer_1

The basic idea of this paper is to replace the MCGSM (mixture of conditional Gaussian scale mixtures) of [38] with a version where a continuous-valued hidden state vector h is maintained in a LSTM. This is used as a model of natural images and assessed by a density estimation task (secs 3.2 and 3.3, Tables 1-3), and for texture synthesis and inpainting (sec 3.4).

The model for p(x_ij|h_ij) (l 150) is in fact not specified at all. Given that h is a continuous-valued vector (es per eq 6) we need to see some functional form.

RNADE [41] is designed for fixed-length vectors. In the case of "infinite dimensional images" it would be very simple to build a mixture density network (MDN) or mixture of experts (see e.g. Bishop's 1995 book sec 6.4) to predict x_{ij} given a finite window of context (as per Fig 1A); (R)NADE is used in the more complex case of fixed-length data vectors. This comparison needs to be made to properly evaluate the benefit of RIDE in the Tables, as the MCGSM (as per eqs 4 and 5) is a relatively simple MDN. RIDE has theoretical benefits of potentially using infinite context as opposed to finite context, cf FIR vs IIR filters, but this theoretical benefit needs to be demonstrated in practice.

RIDE is compared with variants of the MCGSM on "infinite dimensional images", and as per Tables 1 and 2 (RH column) it shows small advantages over MCGSM. However, as per the above para, we need to see comparisons with a MDN which is richer than a MCGSM (e.g. using hidden layers).

The authors claim (l 371) that inpainting has only been done for small image regions until recently (citing [32] from 2015).

However, there is work before this inpainting large regions, see e.g.

Multiple Texture Boltzmann Machines, J. J. Kivinen and Christopher K. I. Williams, Proceedings AISTATS 2012, http://homepages.inf.ed.ac.uk/ckiw/postscript/mtbm.pdf . In that paper the authors consider the obvious evaluation of inpainting, computing the normalized cross correlation (NCC) of the inpainted region with the ground truth; this method should be used to

evaluate the RIDE model quantitatively. Also in the same paper the authors used a texture similarity score (TSS) for assessing the

quality of texture samples.

Quality: The paper makes sense technically, using the spatial LSTM model of [8] to model images.

Clarity: Writing is clear, except for defn of p(x_ij|h_ij) (l 150).

Originality: A relatively straightforward application of the spatial LSTM framework (although getting it all to work well would require a lot of effort unless a good spatial LSTM code base is to hand).

Significance: As per the comments above, we need to see (1) comparison

with a MDN predictor (with careful effort to make it work as well as it can, as you have done for LSTM), and (2) better comparisons for texture synthesis/inpainting.

Overall: Marginally below threshold due to lack of comparisons for density estimation and texture work.

Other points:

l 135 provide name for g units.

l 280 Table 3 Aveage -> Average

l 312. The "dead leaves" model was introduced by Matheron in 1968, and further developed by D. Jeulin, see e.g.

Jeulin, D. (1989). Morphological modeling of images by sequential random functions. Signal Processing 16, 403-431.

Jeulin, D. (1996). Dead leaves models: From space tesselation to random functions. In Advances in Theory and Applications of Random Sets. ed. D. Jeulin. pp. 137-156.

l 359 -- give url for where you obtained Brodatz data.

ADDED AFTER REBUTTAL: =====================

The other reviews and the author response have been read.

The author clarification that the model for p(x_ij|h_ij) is a MCGSM means that the results are making a fair comparison between a MCGSM for p(x_ij|x_{< ij}) and p(x_ij|h_ij).

I have thus raised my score to 6.

I still believe that using a more complex model than a MCGSM (with hidden layers) may lead to benefits and encourage the authors to

pursue this line of enquiry.
Summary: The authors propose using a spatial LSTM (dubbed RIDE) to model natural images/textures. For "infinite dimensional images" the model is shown to outperform MCGSM by a small margin (Tables 1-3), but a comparison with mixture density networks is missing.

RIDE is also used for texture synthesis/inpainting, but here only qualitative results are shown, compared to prior work (see below for more details).

Submitted by Assigned_Reviewer_2

Quality: the method is elegant and powerful. The experiments are heterogeneous, well designed, and well detailed.

Clarity: the paper is very well written.

Originality: while the application of LSTMs for generative visual modeling is not new [10,28,34], the submission presents simple but elegant improvements. The combination of experts with LSTM is new to my knowledge.

Significance: the contribution is significant within the research field of natural image modeling. Apart from the new presented model, no technical or conceptual novelties are presented.
Summary: The paper presents a deep recurrent model for generative image modeling, with promising results in applications such as texture synthesis and in painting. The model is a combination of Spatial Long Short-term Memory (LSTM) and Mixtures of Conditional Gaussian Scale Mixtures (MCGSM).

Submitted by Assigned_Reviewer_3

Standard fully visible approaches to density estimation, model images as a product of conditional distribution over pixels, where each distribution is conditioned on a fixed size context window of anterior pixels. In NADE, these conditional distributions take the form of neural networks with shared parameters, while in MCGSM, these take the form of a mixture of Gaussians. The contribution of this paper is to replace the fix-sized context window used in MCGSM,

with a spatial LSTM, resulting in the RIDE model. A factored version of MCGSM is also introduced in the process to deal with the high-dimensional LSTM outputs.

The model evaluation is thorough and comprises analytical results (log-likelihood) on 3 datasets, as well as qualitative evaluation via sampling and inpainting on the well known Brodatz texture dataset. The proposed model convincingly outperforms the included baseline methods on the van Hateren and dead leaves dataset, matching the performance of a much deeper (and presumably more expensive) model based on learning an inverse diffusion process. Importantly, the authors show that the LSTM performs a rich non-linear processing which cannot be simply emulated with a MCGSMs operating on a larger context window.

Results on BSDS300 are less convincing however. A 2-layer RIDE model is outperformed both by the older RNADE model, as well as the standard (but factored) MCGSM model. RIDE nevertheless outperforms the baseline MCGSM models when moving to a training regime based on "infinite" images, instead of fixed 8x8 patches (details on this process were lacking however). It is regrettable however that the authors did not choose to train RNADEs in this setting, as the BSDS300 dataset is the only experiment where these two models are compared head to head, a comparison which favors RNADE over RIDE.

As a generative model, RIDE clearly yields superior samples to MCGSM, especially on textures featuring long-range dependencies. This is strong evidence for the superiority of the LSTM-based context, though I believe their approach could further benefit from a richer conditional distribution for p(x_ij | h_ij) such as the Mixture Density Network of Graves.

The paper would also benefit from a quantitative analysis of the generated textures, by including the standard Texture Similarity Score (TSS). The authors could then directly compare to [R1] (and the TSS scores referenced there-in), which visually seems to outperform RIDE on texture D103 (very similar to D104).

On the topic of baselines, it is also regrettable that the authors chose not to compare directly to the variational lower-bounds of models featuring latent variables (e.g. DLGM, DRAW).

Detailed feedback: Section 2.3 could benefit from a more detailed exposition. From Fig.1 (right) and having a strong "neural network" prior, I assumed the LSTM latent state was simply the input to a linear regressor to predict x_ij. It was thus not immediately clear how h_ij was combined with the MCGSM (though is clear now in hindsight). Section 3.2: how are the "infinitely large images" generated ? Are the images simply tiled in a torus manner ? How are boundary conditions handled in this setting ? Section 3.4: Could the authors comment on what happened when no curriculum is used ? This seems to be a recurring theme with LSTM-based approaches, and it would be nice to shed more light on this. line 304: typo in "preprocessing" Figure 2 (legend): should this not read RIDE instead of RIM ? Table 3 (caption): typo in "average".

[R1] Texture Modeling with Convolutional Spike-and-Slab RBMs and Deep Extensions. Heng Luo Pierre Luc Carrier Aaron Courville Yoshua Bengio. AISTATS'13.
Summary: The authors replace the fixed context window used in convolutional auto-regressive models, with an LSTM. While the idea is straighforward, I do not believe this approach has been used in the litterature and will no doubt become standard procedure in the future. The experimental section is thorough, though regrettably suffers from missing baselines.

Submitted by Assigned_Reviewer_4

Paper Summary:

This paper proposes a novel approach for modeling statistical dependencies in natural images, employing a (multi-layer) 2D recurrent neural structure (using LSTM hidden units) to predict pixel values conditioned on "previous" pixels (when arranged in scanline order) from a local neighborhood.

When compared to other state-of-the art approaches such as RNADE and MCGSMs, the proposed method shows strong performance (in terms of per-pixel log-likelihood rates) on the tasks evaluated.

Quality:

The authors do a good job at situating the proposed approach relative to related methods from the literature. The derivations and experiments (both in the main paper and supplements) seem thorough (although I did not perform a detailed check of all equations), and the authors seem to have put a lot of effort into making comparisons with their method and similar ones.

One slightly different comparison and/or discussion point that I would be interested to see is with Adversarial Generative Nets. (In particular, the recent work "Deep Generative Image Models Using A Laplacian Pyramid of Adversarial Networks" by Denton et al also shows very strong generative performance. However, this is perhaps an unfair request given the timing of NIPS submissions and the publication date of that work.) A drawback of these models, however, is the inability to obtain direct computations of log-likelihoods.

Lastly, given the apparent scalabilty of this approach and its applicability to multidimensional images (as shown in the supplement), it would be interesting to see this method used on "real" images as opposed to small patches or generated textures.

Clarity:

The paper is well structured and the methods are clearly explained and motivated.

Originality:

The use of 2D LSTM gates in the way employed in this paper is (to my knowledge) a novel contribution, and the method's tractability and scalability are both appealing properties.

Significance:

I am not convinced of the large-scale significance of this work, despite some of these favorable properties. The improvement over related methods/baselines is notable, but not dramatic.
Summary: This paper proposes a novel model for the statistical structure of natural images, and the authors show good performance on some simple benchmark tasks as compared with similar recent approaches. The paper is clear, thorough, and well written. However, the broader significance of this approach was less clear to me.

Author Feedback
Author rebuttal: Thank you for taking the time to review our paper and for suggesting ways to improve it. We hope some of your concerns will already be addressed by our clarifications below, and we would be grateful if you could reconsider your evaluation after taking these into account.

(R1, R4)
"p(x_ij|h_ij) (l 150) is in fact not specified at all."

We used the factorized MCGSM here, which is specified in Equations 3, 4, and 5 (simply replace x_{ < ij} by h_ij). We were hoping this would be clear from Section 2.3 ("The hidden states are then fed into a factorized MCGSM"), but we will update our paper to make it clearer.

(R1)
"RIDE shows small advantages over MCGSM."

Please note that:

- We are already comparing to an improved version of the MCGSM. Thanks to our factorization and conditional whitening, we here are able to train up to 128 components (instead of 8 components, as in Theis et al., 2012).

- When RNADE was first published (NIPS, 2013), their results on BSDS300 were worse than those achieved with GMMs (-0.014 [bit/px]). Deep GMMs (NIPS, 2014) were worse than EoRNADE (-0.018 [bit/px]). This shows that achieving an improvement on this dataset is no trivial task. In light of this, our improvements over an MCGSM with 16 components (closest to model in Theis et al., 2012) of 0.211 [bit/px] are huge.

(R1, R4)
MCGSM, MDN, and RNADE:

RNADE is an MDN with hidden layers. The only difference is a weight sharing scheme which precludes RNADE from being applied to large images directly. Our comparison on 8x8 image patches shows that a factorized MCGSM (even before including hidden layers) is a better parametrization for natural images than this highly optimized MDN (as there have been several papers on RNADE and its precursor). The characterization of the MCGSM as just a "simple MDN" therefore seems a bit unfair.

Please also note that a comparison of RNADE (Uria et al., 2013; 2014) with an MCGSM (Theis et al., 2012) has value in itself, as RNADE was never compared to the MCGSM before. This should be of interest to the NIPS community, which has hosted several papers whose results are included in or which are closely related to our comparison in Table 1.

(R1, R4)
RNADE/MDN for large images:

While it would be fairly easy to extend an MDN ala RNADE to large images, running the experiments and making sure the model is given the best possible chance would take considerable effort. Note that our experiments are already quite comprehensive, as we quantitatively evaluate 2 models (MCGSM and RIDE, not including ensembles) on 3 datasets, and for 2 of these datasets we ran experiments on 2 image sizes (i.e., 10 different experimental settings).

(R1, R4)
"The paper would also benefit from a quantitative analysis of the generated textures [...]"

Note that this paper is primarily about the introduction of a new image model and not its applications. For such papers it is common to show inpainting results as a proof of concept (e.g., Uria et al., 2013; Sohl-Dickstein et al., 2015). In contrast, papers like those of Kivinen & Williams (2012) or Zoran & Weiss (2011) focus on the use of existing models in applications. This separation makes sense, as getting competitive results in these tasks often requires a lot of additional engineering work. Note that we don't claim superior but merely "promising" inpainting performance in our paper. Please also take into account that the space available in a conference paper is quite limited.

(R1)
"The authors claim (l 371) that inpainting has only been done for small image regions [...]"

Will be fixed, thanks.

(R4)
"[...] which visually seems to outperform RIDE on texture D103 [...]"

The textures in [R1] seem to be downsampled, as is common (e.g., Heess, 2009). This would reduce long-range correlations to short-range correlations, which are easier to model.

(R4)
RIDE is outperformed by RNADE/MCGSM on 8x8 image patches:

Note that RNADE suffers a similar drop in performance after including hidden layers:

RNADE: 152.1 [nat] -> RNADE, 1 hl: 143.2 [nat]
MCGSM: 155.8 [nat] -> RIDE, 1 layer: 150.7 [nat]

Instead of spending a lot of engineering effort on getting RIDE to work on the rather cumbersome 63-dimensional dataset, we decided to show what happens when the number of the boundary pixels becomes negligible.

(R4)
Comparison with DRAW/DLGM:

DRAW only models binary data, which is why it interprets RGB values as probabilities and why the DRAW paper does not include lower bounds for SVHN or CIFAR-10.

(R2):
Comparison with GAN/LAPGAN:

The dimensionality of the noise used as input to GAN/LAPGAN is of much lower dimensionality than the images. One consequence of this is that these models only capture a lower-dimensional manifold and have avg. log-likelihood of negative infinity.

(R7)
Comparison with RNADE:

Please see Table 1.